# Overcoming Non-monotonicity in Transducer-based Streaming Generation

**Zhengrui Ma** [1 2]   **Yang Feng** [1 2]   **Min Zhang** [3]

## Abstract

Streaming generation models are utilized across fields, with the Transducer architecture being popular in industrial applications. However, its input-synchronous decoding mechanism presents challenges in tasks requiring non-monotonic alignments, such as simultaneous translation. In this research, we address this issue by integrating Transducer's decoding with the history of input stream via a learnable monotonic attention. Our approach leverages the forward-backward algorithm to infer the posterior probability of alignments between the predictor states and input timestamps, which is then used to estimate the monotonic context representations, thereby avoiding the need to enumerate the exponentially large alignment space during training. Extensive experiments show that our MonoAttn-Transducer effectively handles non-monotonic alignments in streaming scenarios, offering a robust solution for complex generation tasks. Code is available at https://github.com/ictnlp/MonoAttn-Transducer.

## 1. Introduction

Streaming generation is a widely studied problem in fields such as speech recognition (Raffel et al., 2017; Zhang et al., 2020; Seide et al., 2024), simultaneous translation (Cho & Esipova, 2016; Gu et al., 2017; Seamless Communication et al., 2023), and speech synthesis (Ma et al., 2020a; Zhang et al., 2024; Wang et al., 2024). Unlike modern turn-based large language models, streaming models need to start generating the output before the input is completely read. This necessitates a careful balance between generation quality and latency.

[1]Institute of Computing Technology, Chinese Academy of Sciences [2]University of Chinese Academy of Sciences [3]School of Future Science and Engineering, Soochow University. Correspondence to: Zhengrui Ma <mazhengrui21b@ict.ac.cn>, Yang Feng (Corresponding Author) <fengyang@ict.ac.cn>.

*Proceedings of the 42$^{nd}$ International Conference on Machine Learning*, Vancouver, Canada. PMLR 267, 2025. Copyright 2025 by the author(s).

Popular streaming generation methods can be broadly divided into two categories: Attention-based Encoder-Decoder (AED; Bahdanau et al., 2015) and Transducer (Graves, 2012). Streaming AED models adapt the conventional sequence-to-sequence framework (Bahdanau, 2014) to support streaming generation. They often rely on an external policy module to determine the READ/WRITE actions in inference and to direct the scope of cross-attention in training. Examples include Wait-$k$ policy (Ma et al., 2019) and monotonic attention-based methods (Raffel et al., 2017; Arivazhagan et al., 2019; Ma et al., 2020d; 2023a). On the other hand, Transducer models connect the encoder and predictor through a joiner rather than using cross-attention. The joiner is designed to synchronize the encoder and predictor by expanding its output vocabulary to include a blank symbol $\epsilon$, which indicates a READ action. Due to the decoupling of the predictor state from the encoder state, READ/WRITE states in Transducer can be represented by a two-dimensional lattice. This allows for the computation of total probabilities using the forward-backward algorithm (Graves, 2012), facilitating end-to-end optimization. Benefited from joint optimization of all potential policies during training, Transducer often demonstrates better performance compared to AED models (Xue et al., 2022; Wang et al., 2023).

During the decoding process of Transducer, each target token is explicitly aligned with a corresponding source token. This input-synchronous decoding property makes the architecture well-suited for tasks like speech recognition, where the input and output align monotonically. However, it poses challenges for non-monotonic alignment tasks such as simultaneous translation (Chuang et al., 2021; Shao & Feng, 2022; Ma et al., 2023b;c). Due to the decoupled design, Transducer models have limited ability to attend to the input stream history during decoding, making it hard to manage reorderings. To address this issue, recent research (Liu et al., 2021; Tang et al., 2023) has started to explore the incorporation of cross-attention mechanism to enhance the capacity for handling complex non-monotonicity. Despite these efforts, the integration of cross-attention presents significant challenges. By integrating the predictor states with source history through attention, the representation of predictor states becomes relevant not only to the encoder states but also to the specific READ/WRITE path history

(Tang et al., 2023). This results in an exponentially large state space for Transducer, hindering the application of the forward-backward algorithm for end-to-end training.

In this research, we present an efficient training algorithm for Transducer models to learn the monotonic cross-attention mechanism. This allows Transducer's predictor to access source history in real-time inference, improving its ability to handle tasks with non-monotonic alignments. We leverage the forward-backward algorithm to infer the posterior probability of alignments between predictor and encoder states in training. This derived posterior alignment enables the estimation of context representation for each predictor state using expected soft attention. In this way, Transducer models adaptively adjust the scope of attention based on their predictions, avoiding the need to enumerate the exponentially large alignment space during training.

We conduct experiments on both speech-to-text/speech simultaneous translation to demonstrate the generality of our approach across various modalities. MonoAttn-Transducer shows significant improvements in generation quality without a noticeable increase in latency in both *ideal* and *computation-aware* settings (§5). Further analysis reveals that MonoAttn-Transducer is particularly effective in handling samples with higher levels of non-monotonicity (§6).

## 2. Background

### 2.1. Streaming Generation

Streaming generation models typically process a streaming input $x = \{x_1, ..., x_T\}$ and generate a target sequence $y = \{y_1, ..., y_U\}$ in a streaming manner. To measure the amount of source information utilized during generation, a monotonic non-decreasing function $g(u)$ is introduced to represent the number of observed source tokens at the time of generating $y_u$.

### 2.2. Transducer

Transducer model (Graves, 2012) comprises three components: an encoder, a predictor, and a joiner. The encoder unidirectionally encodes the received input prefix $x_{1:t}$ into a context representation $h_t$. The predictor functions similarly to an autoregressive language model, encoding the dependencies between tokens in the generated prefix $y_{1:u}$ into $s_u$. The joiner makes predictions based on the current source representation $h_t$ and target representation $s_u$. If the model needs to READ more information to update the source representation for continued generation, a blank token $\epsilon$ is generated. Otherwise, a WRITE operation is performed, and the generated token is fed back into the predictor to obtain a new target representation. Each time $h_t$ or $s_u$ is updated, the joiner performs a prediction step until the entire source has been processed. The encoder and predictor

are usually modeled using either a recurrent neural network (Graves, 2012) or Transformer layers (Zhang et al., 2020). The joiner is typically composed of a feed-forward network.

Since explicit alignment information for parallel pairs is not available during training, it is necessary to solve for the total probabilities of all READ/WRITE paths that can generate the target to perform maximum likelihood estimation. Given that the state space of Transducer form a two-dimensional lattice, the forward-backward algorithm can be utilized to compute the total probability. Define the forward and backward variables as:

$$
\begin{aligned}
\alpha(t, u) &\coloneqq p(y_{1:u}|x_{1:t}) \\
\beta(t, u) &\coloneqq p(y_{u+1:U}|x_{t:T})
\end{aligned}
\tag{1}
$$

The forward and backward variables for all $1 \leq t \leq T$ and $0 \leq u \leq U$ can be calculated recursively:

$$
\begin{aligned}
\alpha(t, u) \\
= \alpha(t-1, u)p(\epsilon|t-1, u) + \alpha(t, u-1)p(y_u|t, u-1) \\
\beta(t, u) \\
= \beta(t+1, u)p(\epsilon|t, u) + \beta(t, u+1)p(y_{u+1}|t, u)
\end{aligned}
\tag{2}
$$

with initial condition $\alpha(1, 0) = 1$ and $\beta(T, U) = p(\epsilon|T, U)$. $p(v|t, u)$ denotes the probability of generating token $v$ from $h_t$ and $s_u$, $v \in \mathcal{V} \cup \{\epsilon\}$. The total output probability is:

$$
p(\boldsymbol{y}|\boldsymbol{x}) = \alpha(T, U)p(\epsilon|T, U).
\tag{3}
$$

By leveraging the forward-backward algorithm, Transducer models are trained to implicitly acquire the READ/WRITE policy from the data.

## 3. Method

In this section, we provide a detailed introduction to our proposed MonoAttn-Transducer.

### 3.1. Overview

MonoAttn-Transducer works similarly to standard Transducer, with the key difference being that its predictor can attend to the encoder history using monotonic attention. During streaming generation, the scope of monotonic attention includes all source context representations that have already appeared. Formally, when the predictor encodes the $u$-th target state, it depends on the representations of previous target states and the existing source context:

$$
s_u = f_\theta(s_{0:u-1}, h_{1:g(u)}),
\tag{4}
$$

where $1 \leq u \leq U$ and $g(u)$ denotes the number of observed source tokens at the time of generating $y_u$. The edge case $s_0$ is defined as $s_0 = f_\theta(h_1)$. In both Transducer and MonoAttn-Transducer, token $y_u$ is generated based

on source representation $h_{g(u)}$ and target representation $s_{u-1}$. Given $s_{u-1}$ can only attend to source contexts up to $g(u-1)$ through monotonic attention, related information in $x_{g(u-1)+1:g(u)}$ should ideally be encoded within $h_{g(u)}$.

## 3.2. Training Algorithm

Training MonoAttn-Transducer is challenging as it exponentially expands Transducer's state space. To address this issue, we firstly leverage the forward-backward algorithm to compute the posterior probability of aligning target representation $s_u$ with source representation $h_t$ (i.e., the probability of generating token $y_u$ immediately after reading $x_t$). This posterior alignment is then used to estimate the expected context vector in the monotonic cross-attention for each predictor state in training. Detailed explanations are provided in the following.

### 3.2.1. POSTERIOR ALIGNMENT

Suppose we have a probability lattice $p(v|t, u)$, representing the probability of generating token $v$ from $h_t$ and $s_u$, for $1 \le t \le T$, $0 \le u \le U$, and $v \in \mathcal{V} \cup \{\epsilon\}$. The posterior probability of generating $y_u$ at the moment $x_t$ is read can be represented by:

$$\pi_{u,t} = \frac{p(y_{1:u-1}|x_{1:t})p(y_u|t, u-1)p(y_{u+1:U}|x_{t:T})}{p(y_{1:U}|x_{1:T})} \quad (5)$$

with the edge case:

$$\pi_{0,t} = \begin{cases} 1 & t = 1 \\ 0 & t \ne 1 \end{cases} \quad (6)$$

which implies that the predictor state $s_0$ is generated immediately after the first source token arrives. Using the forward and backward variables introduced in Section 2.2, Eq. 5 can be concisely expressed as follows:

$$\pi_{u,t} = \frac{\alpha(t, u-1)p(y_u|t, u-1)\beta(t, u)}{\alpha(T, U)p(\epsilon|T, U)}. \quad (7)$$

This guarantees that the posterior alignment probability for all pairs $(t, u)$ can be solved in $O(TU)$ time using the above forward-backward algorithm, facilitating the calculation of the expected context representation introduced later.

### 3.2.2. MONOTONIC ATTENTION

The incorporation of monotonic attention makes the representation of predictor states relevant to specific READ/WRITE history, leading to a prohibitively large state space for enumerating alignments. Therefore, we turn to estimate the context vector in monotonic attention based on the posterior alignment probability during training. This approach enables the model to adaptively adjust the scope of cross-attention according to its prediction. Consequently,

MonoAttn-Transducer learns a monotonic attention mechanism while maintaining the same time and space complexity as Transducer.

Formally, given the energy $e_{u,t}$ for the pair consisting of encoder state $h_t$ and predictor state $s_u$, as well as the posterior alignment probability $\pi_{u,t}$, the expected context representation $c_u$ for predictor state $s_u$ can be expressed as:

$$c_u = \sum_{t=1}^{T} \pi_{u,t} \sum_{t'=1}^{t} \frac{\exp(e_{u,t'})}{\sum_{t''=1}^{t} \exp(e_{u,t''})} h_{t'}. \quad (8)$$

This indicates that the expected context representation $c_u$ is a weighted sum of context representations under various amount of source information, with the weights given by the posterior alignment probability $\pi_{u,t}$. The nested summation operations in Eq. 8 may lead to an increase in computational complexity. Fortunately, Arivazhagan et al. (2019) suggests that it can be rewritten as:

$$\phi_{u,t} = \sum_{t'=t}^{T} \frac{\pi_{u,t'} \exp(e_{u,t})}{\sum_{t''=1}^{t'} \exp(e_{u,t''})}$$
$$c_u = \sum_{t=1}^{T} \phi_{u,t} h_t \quad (9)$$

Eq. 9 can then be computed efficiently using cumulative sum operations (Arivazhagan et al., 2019). Then in training, Eq. 4 can be estimated as

$$s_u = f_\theta(s_{0:u-1}, c_u). \quad (10)$$

### 3.2.3. TRAINING WITH PRIOR ALIGNMENT

The above algorithm facilitates MonoAttn-Transducer in learning monotonic cross-attention with posterior alignment probability. However, this presents a chicken-and-egg paradox: the posterior alignment is derived from an output probability lattice constructed using an estimated context representation, while the context vector is, in turn, estimated using a posterior alignment. We address this problem by using a prior alignment to construct a prior output probability lattice. This lattice is then used to infer the posterior alignment and train MonoAttn-Transducer's monotonic attention.

There are several options for the prior alignment $p_{u,t}$. The simplest one is the uniform distribution, which assigns an equal probability of being generated at any timestep for all the target tokens:

$$p_{u,t}^{\text{uni}} = \frac{1}{T}, \ 1 \le t \le T, \ 1 \le u \le U. \quad (11)$$

The edge case $p_{0,t}^{\text{uni}}$ is similar to the situation of $\pi_{0,t}$, where all the probability mass is concentrated at $t = 1$.

However, it is preferable to select a more reasonable prior. An ideal prior alignment should ensure that the posterior alignment, derived from the lattice constructed using the prior, can accurately estimate the expected context representation. In streaming generation tasks, even though there may be reorderings in the mapping from source to target, a certain level of monotonic alignment is generally maintained. Therefore, we propose introducing a prior distribution $p_{u,t}^{\text{dia}}$, which assumes that the number of tokens generated for each READ action is uniformly distributed:

$$
\begin{aligned}
w_{u,t} &= \exp\left(-|u - \frac{t \cdot U}{T}|\right) \\
p_{u,t}^{\text{dia}} &= \frac{w_{u,t}}{\sum_{t'=1}^{T} w_{u,t'}}
\end{aligned}
\tag{12}
$$

for $1 \leq t \leq T$, $1 \leq u \leq U$. The edge case $p_{0,t}^{\text{dia}}$ is handled in the same manner as $p_{0,t}^{\text{uni}}$. This prior assumes a uniform mapping between the source and target, such that each target token is most likely generated at the time its corresponding source token is read. The probability decreases as the time difference from this moment increases.

### 3.2.4. Chunk Synchronization

In speech audio, there often exists strong temporal dependencies between adjacent frames. Therefore, a chunk size $C$ is typically set, and the streaming model makes decisions only after receiving a speech chunk (Ma et al., 2020c). In terms of Transducer models, when a READ action is taken, the source representation is updated after a new speech chunk is read. The new source representation is then set as the representation of the last frame in the chunk (Liu et al., 2021; Tang et al., 2023). In such a situation, the receptive field of MonoAttn-Transducer's cross-attention for predictor state $s_u$ encompasses all hidden states in the received chunks, i.e., $h_{1:C \cdot \tilde{g}(u)}$, where $\tilde{g}(u)$ denotes the number of received chunks when generating token $y_u$. To bridge the gap between training and inference, the posterior alignment probability utilized in training process is adjusted by transferring all the probability mass on encoder states within a chunk to the last state in the chunk:

$$
\tilde{\pi}_{u,t} = \begin{cases} \sum_{t'=(d-1)\cdot C+1}^{d \cdot C} \pi_{u,t'} & t = d \cdot C \\ 0 & t \neq d \cdot C \end{cases}
\tag{13}
$$
$$
\text{for} \quad d = 1, 2, 3, \ldots
$$

The prior alignment probability is adjusted in the same manner. We detail the entire training process in Alg. 1.

## 4. Related Work

Our work is closely related to researches in designing cross-attention modules for Transducer models. Prabhavalkar et al. (2017) pioneered the use of attention to link the predictor and encoder. However, their design requires the entire

---

**Algorithm 1** Training Algorithm of MonoAttn-Transducer

**Input**: Source $x$, Target $y$, Chunk Size $C$
**Output**: Training Loss $\mathcal{L}$

1: Compute prior alignment $p_{u,t}^{\text{dia}}$ (Eq. 12)
2: Compute chunk-synchronized prior alignment $\tilde{p}_{u,t}^{\text{dia}}$ based on chunk size $C$ (Eq. 13)
3: Estimate context $c_u^{\text{prior}}$ with $\tilde{p}_{u,t}^{\text{dia}}$ (Eq. 9)
4: Forward MonoAttn-Transducer with $c_u^{\text{prior}}$
5: Infer posterior alignment $\pi_{u,t}$ (Eq. 7)
6: Compute chunk-synchronized posterior alignment $\tilde{\pi}_{u,t}$ based on chunk size $C$ (Eq. 13)
7: Estimate context $c_u$ with $\tilde{\pi}_{u,t}$ (Eq. 9)
8: Forward MonoAttn-Transducer with $c_u$
9: Calculate total output probability $\mathcal{L}$ (Eq. 3)
10: **return** $\mathcal{L}$

---

source to be available, limiting it to offline generation. For streaming generation, the receptive field of attention must synchronize with the input. This synchronization leads to an exponentially large state space, which significantly complicates the training process. To mitigate this issue, Liu et al. (2021) separated the predictor's cross-attention from its self-attention, ensuring that cross-attention occurs only after self-attention. This approach maintains the independence of predictor states from READ/WRITE path history, allowing for standard training methods. However, this separation limits the richness of the predictor's learned representations. Alternatively, Tang et al. (2023) proposed updating the representation of all predictor states whenever a new source token is received. While this method also preserves the independence of predictor states from READ/WRITE path history, it significantly increases both inference-time computational complexity and training-time memory requirements. It necessitates an additional $(T-1)$ forward passes of the predictor during decoding, which adversely affects latency-sensitive streaming generation. Furthermore, the GPU memory usage for attention during training increases from $O(1)$ to $O(T)$, leading to prohibitively high training costs and limiting the model's scalability. In contrast to the above, the proposed MonoAttn-Transducer maintains the same time complexity and memory overhead as Transducer. A detailed comparison between these methods is summarized in Table 1.

Our work is also related to researches in designing attention modules for streaming AED models. These works often introduce Bernoulli variables to indicate READ/WRITE actions. The distribution of these variables is used to estimate monotonic alignment and to compute the expected context representation in training (Raffel et al., 2017). Depending on the setting of attention window, these works can be classified into monotonic hard attention (Raffel et al., 2017), monotonic chunkwise attention (MoChA; Chiu & Raffel,

*Table 1.* Comparison of Transducer-based streaming models. *Computational Complexity* refers to the number of forward passes executed by the predictor in inference. *Memory Overhead* refers to the memory consumption of the attention module in training.

| Method | Merge Module | Computational Complexity | Memory Overhead |
|---|---|---|---|
| Transducer (Graves, 2012) | Joiner | $O(U)$ | N/A |
| CAAT (Liu et al., 2021) | Joiner | $O(U)$ | $O(T)$ |
| TAED (Tang et al., 2023) | Predictor, Joiner | $O(U + T)$ | $O(T)$ |
| MonoAttn-Transducer (Ours) | Predictor, Joiner | $O(U)$ | $O(1)$ |

2018), and monotonic infinite lookback attention (MILk; Arivazhagan et al., 2019). Ma et al. (2020d) subsequently introduced the MILk mechanism to Transformer models, and Ma et al. (2023a) further proposed a numerically-stable algorithm for estimating monotonic alignment. Unlike the aforementioned works, our approach learns monotonic attention based on the posterior alignment of Transducer, avoiding the use of unstable Bernoulli variables.

## 5. Experiments

We validate the performance of our MonoAttn-Transducer on two typical streaming generation tasks: speech-to-text and speech-to-speech simultaneous translation. The differences in grammatical structures between the source and target languages often necessitate word reordering during generating translation. This property makes the simultaneous translation task well-suited for evaluating the ability in handling non-monotonic alignments.

### 5.1. Experimental Setup

**Datasets** We conduct experiments on two language pairs of MuST-C speech-to-text translation datasets: English to German (En→De) and English to Spanish (En→Es) (Di Gangi et al., 2019). For speech-to-speech experiments, we evaluate models on CVSS-C French to English (Fr→En) dataset (Jia et al., 2022).

**Model Configuration** We use the open-source implementation of Transformer-Transducer (Zhang et al., 2020) from Liu et al. (2021) as baseline and build our MonoAttn-Transducer upon it. The speech encoder consists of two layers of causal 2D-convolution followed by 16 chunk-wise Transformer layers with pre-norm. Each convolution layer has a 3×3 kernel with 64 channels and a stride size of 2, resulting in a downsampling ratio of 4. In chunk-wise Transformer layers, the speech encoder can access states from all previous chunks and one chunk ahead of the current chunk (Wu et al., 2020; Shi et al., 2021). The chunk size is adjusted within the set {320, 640, 960, 1280}ms. Offline results are obtained by setting the chunk size longer than any utterance in the corpus. Both sinusoidal positional encoding (Vaswani et al., 2017) and relative positional attention (Shaw et al.,

2018) are incorporated into the speech encoder. Sinusoidal positional encoding is applied after the convolution layers. The predictor comprises two autoregressive Transformer layers with post-norm, utilizing only sinusoidal positional encoding. The monotonic attention is similar to standard cross-attention but differs in its receptive field. The joiner is implemented as a simple FFN. We incorporate the multi-step decision mechanism (Liu et al., 2021) with a decision step of 4. All Transformer layers described above are configured with a 512 embedding dimension, 8 attention heads and a 2048 FFN dimension. The total number of parameters for the Transducer baseline and MonoAttn-Transducer are 65M and 67M, respectively. More implementation details are provided in App. A.

**Evaluation** We use SimulEval toolkit (Ma et al., 2020b) for evaluation. Translation quality is assessed using case-sensitive detokenized `BLEU` (Papineni et al., 2002; Post, 2018) and neural-based `COMET-22` score. Latency is measured by word-level Average Lagging (`AL`; Ma et al., 2019; 2020c) and Length-Adaptive Average Lagging (`LAAL`; Papi et al., 2022).[1] For speech-to-speech tasks, translation quality is assessed using `ASR-BLEU` and latency is measured by delay of generated waveform chunks (Ma et al., 2022).

### 5.2. Main Results

We evaluate the performance of MonoAttn-Transducer against Transducer baseline across various latency conditions obtained by varying the chunk size. In this comparison, we consider two configurations of MonoAttn-Transducer. The first, referred to as MonoAttn-Transducer-*Posterior*, is trained strictly according to Algorithm 1. The second, termed MonoAttn-Transducer-*Prior*, is optimized directly using prior alignment, without inferring the posterior (calculate total output probability $\mathcal{L}$ using $c_u^{\mathrm{prior}}$). Results are shown in Table 2.

It can be observed that MonoAttn-Transducer-*Posterior* significantly outperforms the Transducer baseline across var-

---

[1]Numerical results with more metrics are provided in App. C. Notably, Table 6 presents a comparison of the *computation-aware* latency metrics for `AL` and `LAAL` between the Transducer and MonoAttn-Transducer models.

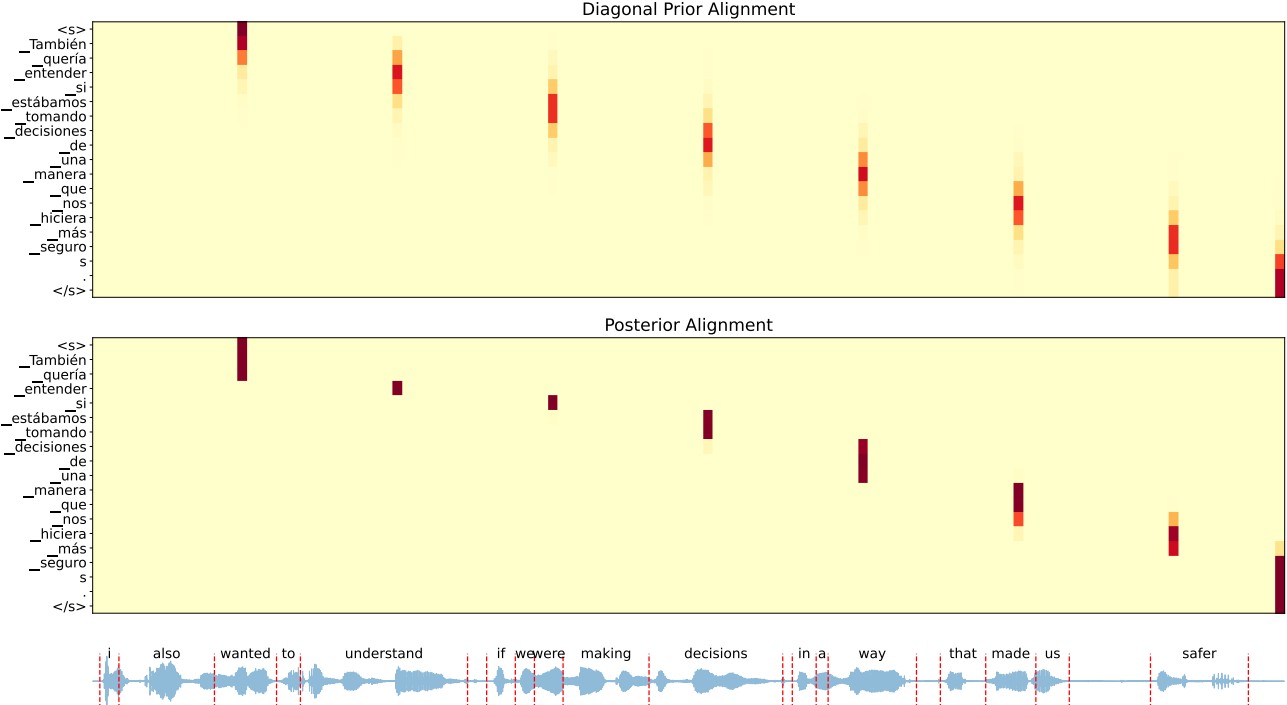

Figure 1. An example of diagonal prior and posterior alignment from MuST-C English-to-Spanish training corpus. The vertical axis represents the target subword sequence and the horizontal axis represents the speech waveform. Darker areas indicate higher alignment probabilities. Chunk size in this example is set to 640ms. More examples are provided in App. D.

ious settings of chunk size in both translation directions. Specifically, in En-Es, it shows an average improvement of 0.75 BLEU or 0.95 COMET score in generation quality under different latency conditions. In En-De, it achieves an even more significant improvement, with an average increase of as much as 2.06 COMET score, while latency remains nearly unchanged. Further analysis reveals that the benefits of learning monotonic attention are more pronounced with a larger chunk size. Notably, in scenarios where latency exceeds 1.5s and during offline generation, the average improvement reaches 0.88 BLEU or 1.77 COMET score. This can be attributed to MonoAttn-Transducer benefiting more from monotonic attention to handle reorderings when it has flexibility to wait for more source information.

Moreover, we have observed some notable results of MonoAttn-Transducer-*Prior*. With a larger chunk size, the performance of MonoAttn-Transducer-*Prior* is comparable to that of MonoAttn-Transducer-*Posterior*, and even slightly outperforming the latter in En-De. However, there exists a significant performance drop with a smaller chunk size. Specifically, with a chunk size of 320ms, MonoAttn-Transducer-*Prior*'s generation quality is on average 1.03 BLEU lower than Transducer baseline under similar latency conditions. This phenomenon highlights the importance of learning monotonic attention through inferring posterior

alignment. From the chunk synchronization mechanism described in Eq. 13, smaller chunk sizes require finer alignment granularity between the predictor and encoder states. This increased granularity necessitates more precise alignment to estimate the expected context representation during training. Figure 1 provides an example of diagonal prior and posterior alignment. While the diagonal prior generally captures the trend of the alignment information, it can be skewed by the uneven distribution of speech information and possible local reorderings. In contrast, the inferred posterior offers a more confident and accurate alignment probability. For instance, the diagonal prior assigns a high probability to aligning the word *"si (if)"* with the timestep preceding the waveform of *"if"*, while the inferred posterior corrects this misalignment. Therefore, learning monotonic attention with posterior alignment leads to a more accurate estimation of context representation and improved performance.[2] In subsequent experiments, we represent MonoAttn-Transducer using the results of MonoAttn-Transducer-*Posterior*.

---

[2]We present a comparison between the prior and posterior under various chunk sizes in App. D. A key observation is that as the alignment granularity becomes finer, the differences gradually increases.

*Table 2.* Comparison of MonoAttn-Transducer and Transducer across various chunk size settings on MuST-C English to German and English to Spanish datasets.

| | | En-Es | | | | | En-De | | | | |
|---|---|---|---|---|---|---|---|---|---|---|---|
| | **Chunk Size** ($ms$) | 320 | 640 | 960 | 1280 | $\infty$ | 320 | 640 | 960 | 1280 | $\infty$ |
| Transducer | **LAAL** ($ms$) | 1168 | 1466 | 1847 | 2220 | - | 1258 | 1563 | 1942 | 2312 | - |
| | **BLEU** ($\uparrow$) | 24.33 | 25.82 | 26.36 | 26.40 | 26.75 | 19.99 | 22.10 | 22.20 | 22.96 | 23.10 |
| | **COMET** ($\uparrow$) | 67.94 | 69.92 | 70.48 | 70.65 | 71.14 | 62.81 | 65.01 | 65.75 | 66.26 | 67.03 |
| MonoAttn-Transducer (*Posterior*) | **LAAL** ($ms$) | 1230 | 1475 | 1837 | 2204 | - | 1317 | 1582 | 1957 | 2305 | - |
| | **BLEU** ($\uparrow$) | 24.72 | 26.74 | 27.05 | 27.41 | 27.48 | 20.22 | 22.47 | 22.94 | 23.74 | 24.42 |
| | **COMET** ($\uparrow$) | 68.98 | 70.71 | 71.21 | 71.90 | 72.24 | 64.24 | 67.06 | 68.22 | 68.54 | 69.82 |
| MonoAttn-Transducer (*Prior*) | **LAAL** ($ms$) | 1123 | 1419 | 1815 | 2184 | - | 1231 | 1535 | 1929 | 2286 | - |
| | **BLEU** ($\uparrow$) | 23.00 | 26.46 | 27.07 | 27.42 | 27.48 | 19.26 | 22.62 | 23.51 | 24.01 | 24.42 |
| | **COMET** ($\uparrow$) | 68.24 | 70.45 | 71.33 | 71.99 | 72.24 | 63.85 | 67.63 | 68.65 | 69.27 | 69.82 |

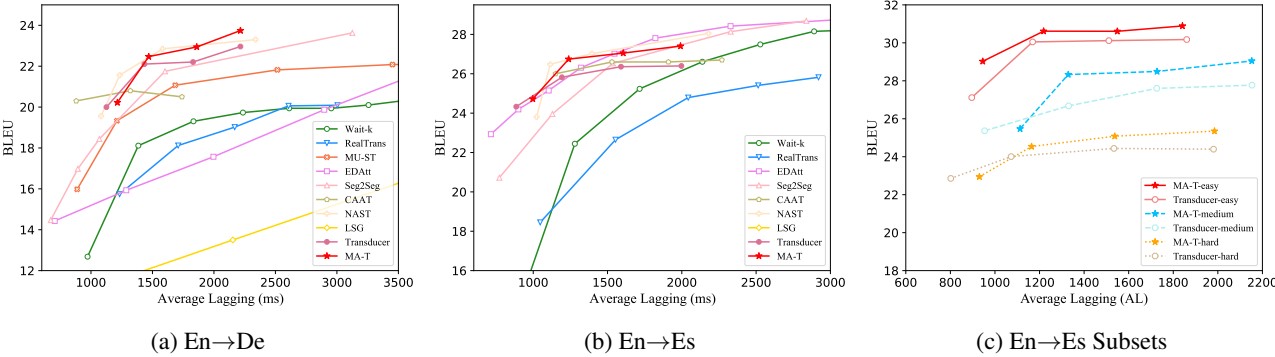

(a) En→De      (b) En→Es      (c) En→Es Subsets

*Figure 2.* **(a)**, **(b)**: Results of translation quality (`BLEU`) against latency (Average Lagging, `AL`) on MuST-C English to German and English to Spanish datasets. **(c)**: Performance on MuST-C English to Spanish test subsets categorized by non-monotonicity. In the figures above, *MA-T* denotes MonoAttn-Transducer.

### 5.3. Comparison with State-of-the-Art

We compare MonoAttn-Transducer with state-of-the-art open-source approaches in simultaneous translation, including Wait-$k$ (Ma et al., 2020c), RealTrans (Zeng et al., 2021), CAAT (Liu et al., 2021), MU-ST (Zhang et al., 2022), EDAtt (Papi et al., 2023a), AlignAtt (Papi et al., 2023b), Seg2Seg (Zhang & Feng, 2023), NAST (Ma et al., 2024) and LLM-based LSG (Guo et al., 2025). Further details about baselines are available in App. B. Results are plotted in Figure 2a and 2b. We observe that learning monotonic attention significantly enhances the performance of Transducer, achieving leading streaming generation quality. Compared to CAAT, another Transducer-based model, MonoAttn-Transducer demonstrates superiority in scenarios with less stringent latency requirements. This clearly demonstrates the advantage of MonoAttn-Transducer's tightly coupled self-attention and cross-attention modules in the predictor, which facilitates the learning of richer representations.

TAED is another Transducer-based model highly relevant to

our work. However, the code and distilled data used to train TAED in Tang et al. (2023) have not been made publicly available. This lack of open access hinders a fair comparison of TAED with our MonoAttn-Transducer. Despite this, we attempt to analyze the performance by comparing each with Transducer baseline in their respective experimental settings. The comparison is shown in Table 7 in App. C. We have observed that the improvement from TAED is more pronounced with smaller chunk sizes, which contrasts with the results of MonoAttn-Transducer. We speculate that this is because, in TAED, the representations of all generated predictor states are updated every time the encoder receives a new speech chunk. This helps TAED generate more accurate representations when the chunk size is small. However, this mechanism in TAED incurs an $O(T + U)$ forward propagation cost during simultaneous inference, which can significantly increase latency in practice due to heavy computational overhead when the chunk size is small. In contrast, MonoAttn-Transducer maintains an $O(U)$ complexity as Transducer baseline. As shown in Table 6, this property

*Table 3.* Performance on CVSS-C French to English speech-to-speech translation.

| | Chunk Size ($ms$) | 320 | Offline |
|---|---|---|---|
| Transducer | **ASR-BLEU** ($\uparrow$) | 17.1 | 18.0 |
| | **LAAL** ($ms$) | 984 | - |
| | **StartOffset** ($ms$) | 1520 | - |
| MonoAttn-Transducer | **ASR-BLEU** ($\uparrow$) | 18.3 | 19.3 |
| | **LAAL** ($ms$) | 918 | - |
| | **StartOffset** ($ms$) | 1491 | - |

minimizes the gap between ideal and computation-aware latency, offering advantages in real-time applications.

### 5.4. Results of Speech Generation

Speech-to-speech simultaneous translation requires implicitly performing ASR, MT and TTS simultaneously, and also handling the non-monotonic alignments between languages, making it suitable to evaluate models on streaming speech generation. We adopted a *textless* setup in our experiments, directly modeling the mapping between speech (Zhao et al., 2024). Results are provided in Table 3.

The results demonstrate that MonoAttn-Transducer significantly reduces generation latency (AL). With a chunk size of 320ms, it achieves Transducer's offline generation quality, but reducing lagging to 118ms. For offline settings, our approaches further improves speech generation quality (19.3 vs. 18.0). These results highlight the effectiveness of our approach in achieving a better quality-latency trade-off also for streaming speech generation.

## 6. Analysis

### 6.1. Handling Non-monotonicity

To illustrate MonoAttn-Transducer's capability in handling reorderings through learning monotonic attention, we evaluate its performance against the Transducer baseline across samples with varying levels of non-monotonicity. Intuitively, samples with a higher number of crosses in the alignments between source transcription and reference text pose greater challenges. We therefore evenly partition the test set based on the number of cross-alignments, labeling them as easy, medium and hard.[3] The results are presented in Figure 2c. We observe that MonoAttn-Transducer shows a more substantial improvement over Transducer in the medium and hard subsets across most chunk size settings. However, with a chunk size of 320ms, the improvement is particularly notable in the easy subset. These findings

---

[3]The easy subset includes samples with a cross count of 1 or fewer. The medium subset contains samples with a cross count between 2 and 6. Samples with a cross count greater than 6 are classified as hard.

highlight the unique capabilities of MonoAttn-Transducer in managing non-monotonic alignments. As analyzed in Section 5.2, MonoAttn-Transducer benefits more from learning monotonic attention with a larger chunk size, and this enhanced ability is evident in subsets with higher levels of non-monotonicity. On the other hand, when the chunk size is extremely small, MonoAttn-Transducer has limited flexibility to wait for more source information before processing, thus showing more significant improvement in the easy subset under the condition.

### 6.2. Training Efficiency

**Training Time**: We analyze each step in Algorithm 1 to compare the training time differences between MonoAttn-Transducer and baseline. We observe that Lines 1, 2, 6 involve naive matrix computation without requiring gradients. The additional time overhead introduced by our method arises from Lines 3, 4, 5. Specifically, this includes an additional forward pass of the predictor and the computation for the posterior alignment. The overhead from the posterior calculation is approximately equivalent to that incurred during loss calculation, as both rely on the forward-backward algorithm. Empirically, we found MonoAttn-Trasducer is 1.33 times slower than Transducer baseline with the same configuration on Nvidia L40 GPU.

**Memory Consumption**: Compared to baseline, the additional memory overhead of MonoAttn-Transducer comes solely from its monotonic attention module. The extra forward pass of the predictor is performed without requiring gradients, so it is excluded from the computation graph. Empirically, we observed that the peak memory usage of Transducer baseline is 28GB, while MonoAttn-Transducer exhibits a slightly higher peak usage of 32GB when the total number of source frames is fixed at 40,000 on a single Nvidia L40 GPU.

## 7. Conclusion

In this paper, we propose an efficient algorithm to handle the non-monotonicity problem in Transducer-based streaming generation. Extensive experiments demonstrate that our MonoAttn-Transducer significantly improves the ability in handling non-monotonic alignments in streaming generation, offering a robust solution for Transducer-based frameworks to tackle more complex streaming generation tasks.

## Impact Statement

This paper presents work whose goal is to advance the field of Machine Learning. There are many potential societal consequences of our work, none which we feel must be specifically highlighted here.

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

# A. Implementation Details

**Pre-processing** The input speech is represented as 80-dimensional log mel-filterbank coefficients computed every 10ms with a 25ms window. Global channel mean and variance normalization is applied to the input speech. During training, SpecAugment (Park et al., 2019) data augmentation with the LB policy is additionally employed. We use SentencePiece (Kudo & Richardson, 2018) to generate a unigram vocabulary of size 10000 for the source and target text jointly. Sequence-level knowledge distillation (Kim & Rush, 2016) is applied for fair comparison (Liu et al., 2021). For speech-to-speech experiments, we resample the source audio to 16kHz and apply identical preprocessing steps as those used in speech-to-text experiments. For the target speech, we also downsample the audio and extract discrete units utilizing the publicly available pre-trained mHuBERT model and K-means quantizer.[4] No training data manipulation is applied in speech-to-speech experiments.

**Training Details** Considering that training MonoAttn-Transducer involves two critical processes: inferring the posterior alignment and estimating the context vector, instability in either step can lead to training failure. Therefore, we introduce a curriculum learning strategy for MonoAttn-Transducer. We first pretrain the model in an offline setting. In pretraining, all predictor states can attend to the complete source input, and the model is trained as an offline Transducer. This pretraining phase allows the monotonic attention module to warm up by learning full-sentence attention, thereby enhancing its stability during subsequent adaptation to a streaming scenario. In finetuning, we apply Algorithm 1 to adjust MonoAttn-Transducer with various chunk size configurations. During both training phases, we set the dropout rate to 0.1, weight decay to 0.01, and clip gradient norms exceeding 5.0. The dropout rates for activation and attention are both set to 0.1. The pretraining spans $50k$ updates with a batch size of $160k$ tokens. The learning rate gradually warms up to 5e-4 within $4k$ steps. Finetuning involves training for $20k$ updates and other hyper-parameters remain consistent. Throughout the training, we optimize models using the Adam optimizer (Kingma & Ba, 2015). Automatic mixed precision training is applied. It takes approximately one day to pretrain in an offline setting and another day for streaming adaptation on a server with 4 Nvidia L40 GPUs.

# B. Baselines

We compare our proposed MonoAttn-Transducer with the following state-of-the-art open-source approaches (without using pretrained encoder or any data augmentation method for fair comparison).

### AED-based Models

**Wait-$k$** (Ma et al., 2020c): It executes wait-$k$ policy (Ma et al., 2019) by setting the pre-decision window size to 280 ms.

**RealTrans** (Zeng et al., 2021): It detects word number in the streaming speech by counting blanks in CTC transcription and applies wait-$k$-stride-$n$ strategy accordingly.

**MU-ST** (Zhang et al., 2022): It trains an external segmentation model, which is then utilized to detect meaningful units for guiding generation.

**Seg2Seg** (Zhang & Feng, 2023): It alternates between waiting for a source segment and generating a target segment in an autoregressive manner.

**EDAtt** (Papi et al., 2023a): It calculates the attention scores towards the latest received frames of speech, serving as guidance for an offline-trained translation model during simultaneous inference.

**AlignAtt** (Papi et al., 2023b): It exploits the attention information to generate source-target alignments that guide the model during inference. AlignAtt demonstrates superior performance for delays exceeding 2 seconds, whereas our study primarily focuses on scenarios involving delays shorter than 2 seconds.

---

[4]https://github.com/facebookresearch/fairseq/blob/main/examples/speech_to_speech/docs/textless_s2st_real_data.md

**CTC-based Models**

**NAST** (Ma et al., 2024): It introduces a streaming generation model with fast computation speed by leveraging a non-autoregressive transformer and CTC decoding (Graves et al., 2006).

**Transducer-based Models**

**Transducer**: It adopts the standard Transducer framework (Graves, 2012) and utilizes Transformer as its backend network (Zhang et al., 2020).

**CAAT** (Liu et al., 2021): It incorporates a cross-attention module within Transducer's joiner to alleviate its strong monotonic constraint.

**LLM-based Models**

**LSG** (Guo et al., 2025): It enables the LLM to devise a streaming generation policy that balances latency and generation quality by referring a baseline policy.

## C. Numerical Results

In addition to Average Lagging (`AL`; Ma et al., 2020c), we also incorporate Average Proportion (`AP`; Cho & Esipova, 2016), Differentiable Average Lagging (`DAL`; Arivazhagan et al., 2019) and Length-Adaptive Average Lagging (`LAAL`; Papi et al., 2022) as metrics to evaluate the latency. `AL`, `DAL` and `LAAL` are all reported with milliseconds. The trade-off between latency and translation quality is attained by adjusting the chunk size $C$. The offline results are obtained by setting the chunk size to be longer than any utterance in the dataset ($C = \infty$). We use SimulEval `v1.1.4` for evaluation in all the experiments. The numerical results of MonoAttn-Transducer are presented in Table 4 and 5. A comparison of the *computation-aware* latency metrics for `AL` and `LAAL` between the Transducer and MonoAttn-Transducer models is presented in Table 6.

*Table 4.* Numerical results of MonoAttn-Transducer on MuST-C English to German dataset.

| | **MonoAttn-Transducer on En→De** | | | | |
|---|---|---|---|---|---|
| $C(ms)$ | **AP** | **AL** | **DAL** | **LAAL** | **BLEU** |
| 320 | 0.67 | 1215 | 1497 | 1317 | 20.22 |
| 640 | 0.77 | 1470 | 1872 | 1582 | 22.47 |
| 960 | 0.83 | 1860 | 2309 | 1957 | 22.94 |
| 1280 | 0.86 | 2215 | 2719 | 2305 | 23.74 |
| $\infty$ | - | - | - | - | 24.42 |

*Table 5.* Numerical results of MonoAttn-Transducer on MuST-C English to Spanish dataset.

| | **MonoAttn-Transducer on En→Es** | | | | |
|---|---|---|---|---|---|
| $C(ms)$ | **AP** | **AL** | **DAL** | **LAAL** | **BLEU** |
| 320 | 0.74 | 997 | 1534 | 1230 | 24.72 |
| 640 | 0.81 | 1239 | 1854 | 1475 | 26.74 |
| 960 | 0.88 | 1606 | 2304 | 1837 | 27.05 |
| 1280 | 0.93 | 1991 | 2725 | 2204 | 27.41 |
| $\infty$ | - | - | - | - | 27.48 |

*Table 6.* Comparison of MonoAttn-Transducer and Transducer across various chunk size settings on MuST-C English to German and English to Spanish datasets.

| | | En-Es | | | | En-De | | | |
|---|---|---|---|---|---|---|---|---|---|
| | **Chunk Size** ($ms$) | 320 | 640 | 960 | 1280 | 320 | 640 | 960 | 1280 |
| Transducer | **AL** ($ms,\downarrow$) | 886 | 1193 | 1591 | 1997 | 1126 | 1434 | 1830 | 2215 |
| | **AL_CA** ($ms,\downarrow$) | 1121 | 1330 | 1699 | 2085 | 1323 | 1551 | 1920 | 2296 |
| | **LAAL** ($ms,\downarrow$) | 1168 | 1466 | 1847 | 2220 | 1258 | 1563 | 1942 | 2312 |
| | **LAAL_CA** ($ms,\downarrow$) | 1381 | 1589 | 1944 | 2300 | 1444 | 1673 | 2028 | 2389 |
| MonoAttn-Transducer | **AL** ($ms,\downarrow$) | 997 | 1239 | 1606 | 1991 | 1215 | 1470 | 1860 | 2215 |
| | **AL_CA** ($ms,\downarrow$) | 1239 | 1385 | 1724 | 2089 | 1407 | 1596 | 1964 | 2301 |
| | **LAAL** ($ms,\downarrow$) | 1230 | 1475 | 1837 | 2204 | 1317 | 1582 | 1957 | 2305 |
| | **LAAL_CA** ($ms,\downarrow$) | 1453 | 1607 | 1945 | 2295 | 1501 | 1702 | 2056 | 2387 |

*Table 7.* Comparison of results reported in Tang et al. (2023) and our work on MuST-C English to German dataset.

| | **Chunk Size** ($ms$) | 160 | 320 | 480 | 640 |
|---|---|---|---|---|---|
| Transducer (Tang et al., 2023) | **BLEU** ($\uparrow$) | 20.76 | 21.80 | 22.52 | 23.32 |
| | **AL** ($ms,\downarrow$) | 1282 | 1252 | 1306 | 1498 |
| TAED (Tang et al., 2023) | **BLEU** ($\uparrow$) | 21.57 | 22.63 | 23.48 | 23.47 |
| | **AL** ($ms,\downarrow$) | 1263 | 1354 | 1369 | 1903 |
| | **Chunk Size** ($ms$) | 320 | 640 | 960 | 1280 |
| Transducer (Our implementation) | **BLEU** ($\uparrow$) | 19.99 | 22.10 | 22.20 | 22.96 |
| | **AL** ($ms,\downarrow$) | 1126 | 1434 | 1830 | 2215 |
| MonoAttn-Transducer | **BLEU** ($\uparrow$) | 20.22 | 22.47 | 22.94 | 23.74 |
| | **AL** ($ms,\downarrow$) | 1215 | 1470 | 1860 | 2215 |

# D. Visualization

In this section, we present more examples of *diagonal prior* and its posterior from training corpus. We have observed that, even with significant differences in the prior distribution, the posterior remains fairly robust when the chunk size is constant. The vertical axis represents the target subword sequence and the horizontal axis represents the speech waveform. Darker areas indicate higher alignment probabilities. We use Montreal Forced Alignment tools (McAuliffe et al., 2017) to obtain speech-transcription alignments for illustration.

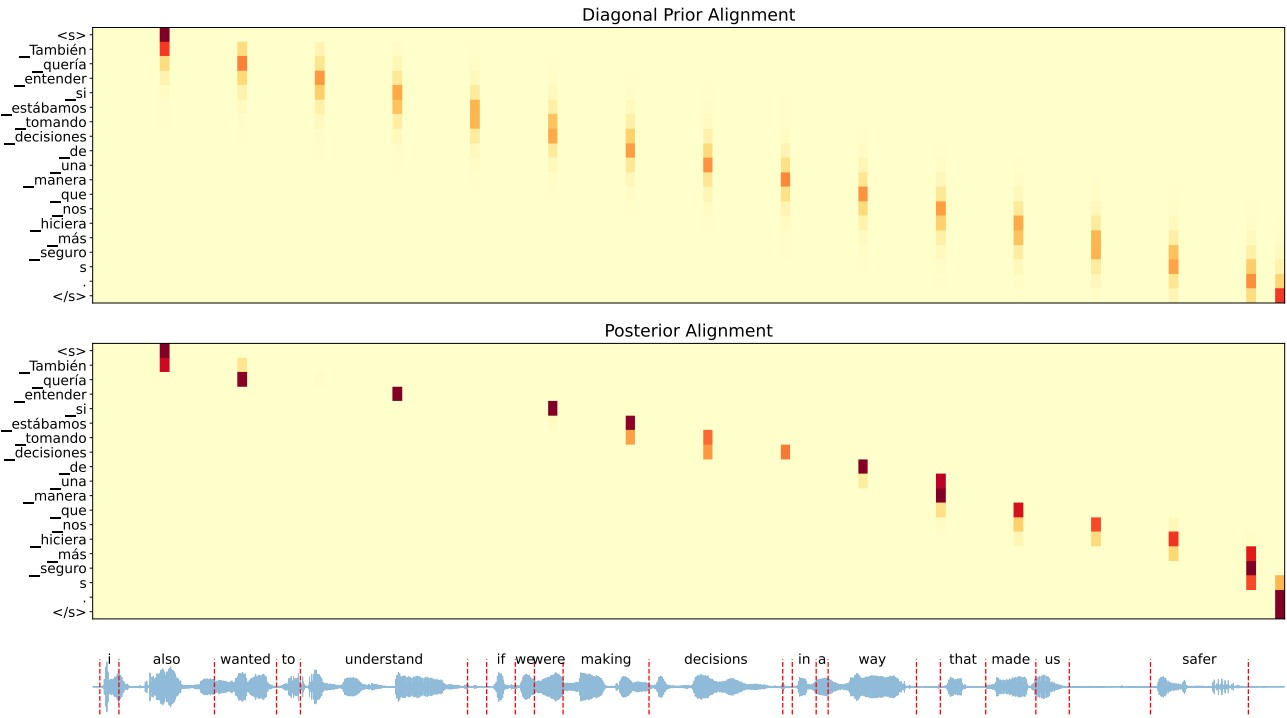

*Figure 3.* Chunk size in this example is set to 320ms. (*Diagonal Prior*)

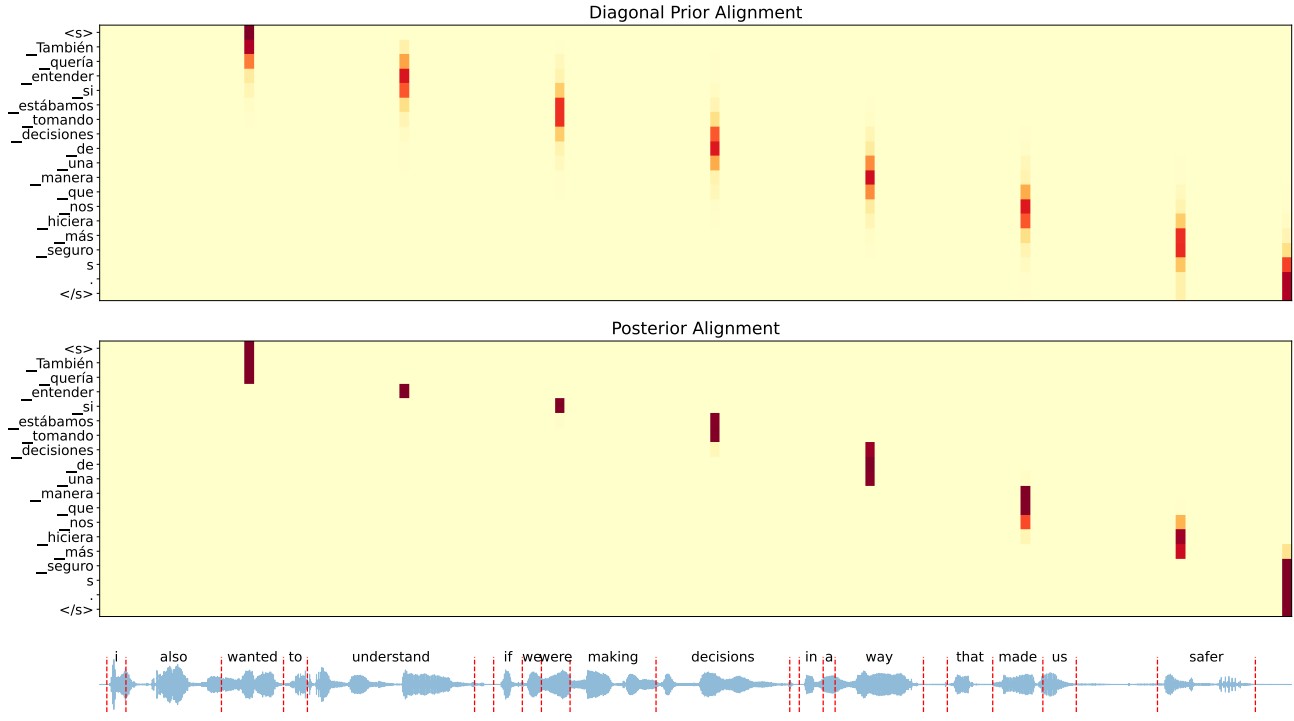

*Figure 4.* Chunk size in this example is set to 640ms. (*Diagonal Prior*)

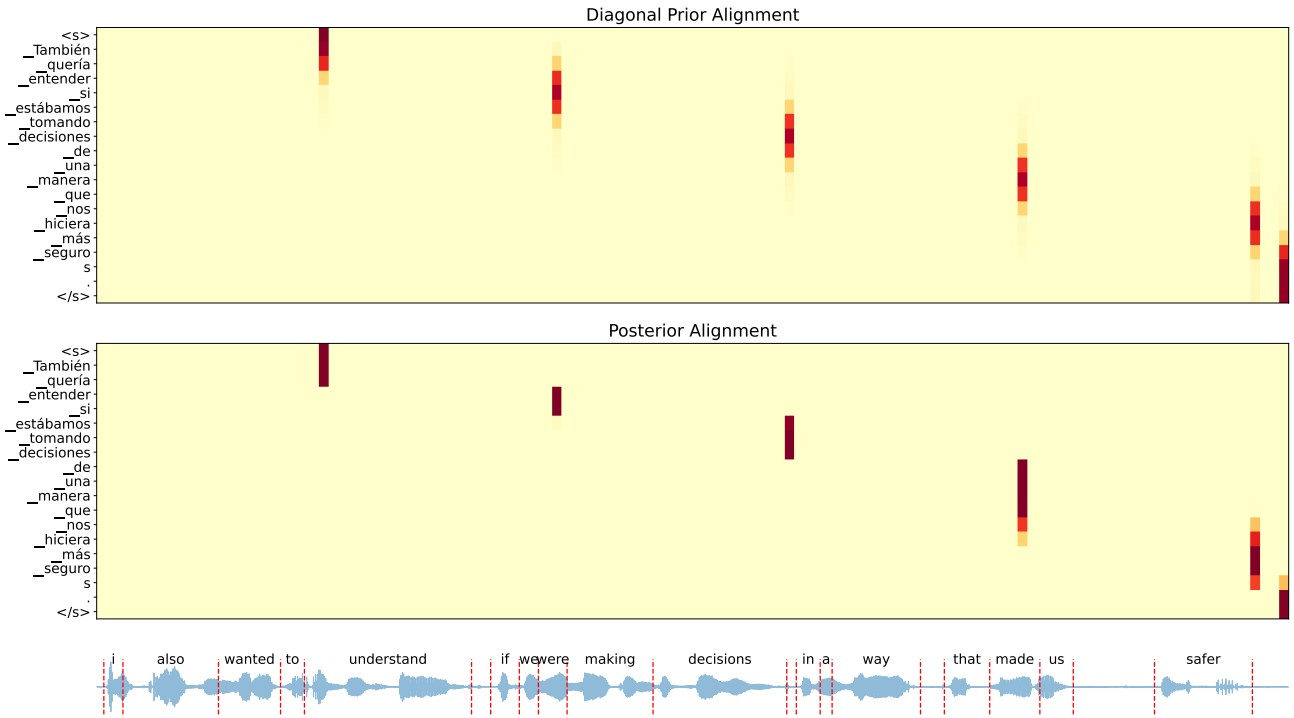

*Figure 5.* Chunk size in this example is set to 960ms. (*Diagonal Prior*)

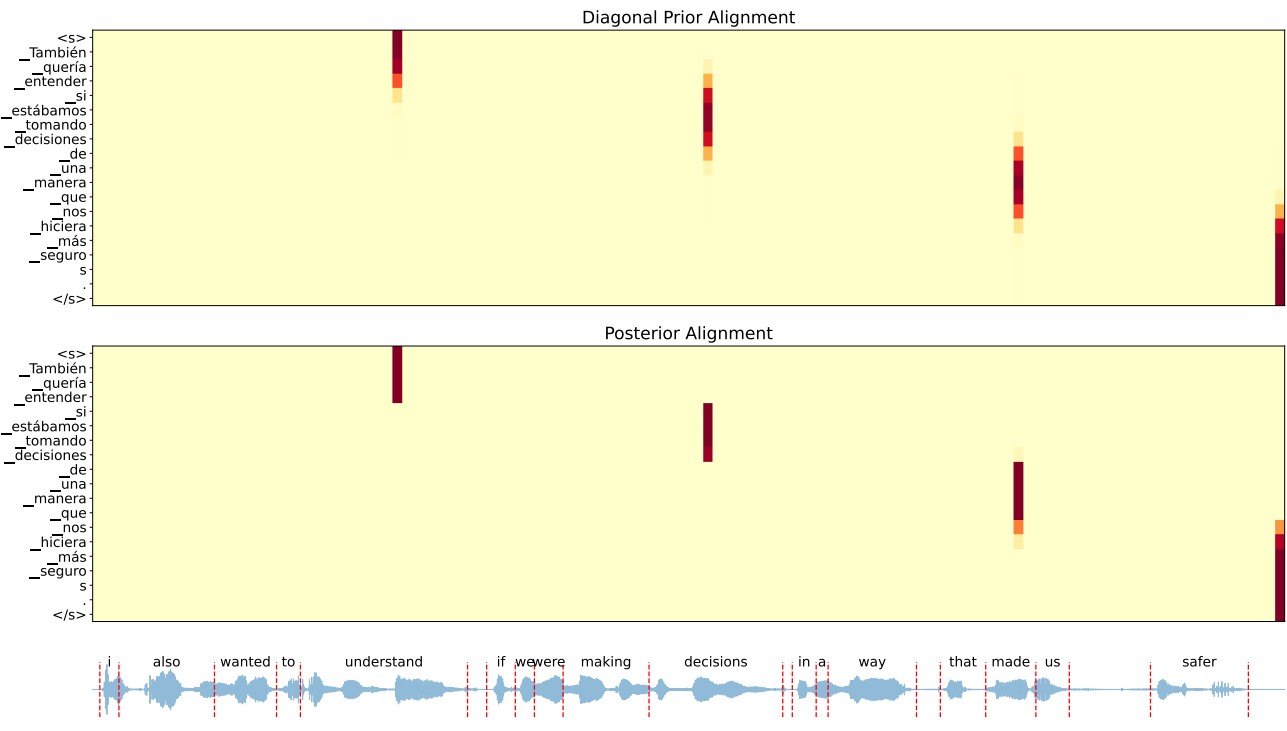

*Figure 6.* Chunk size in this example is set to 1280ms. (*Diagonal Prior*)

