# OpenReview forum: "Overcoming Non-monotonicity in Transducer-based Streaming Generation"
_ICML.cc/2025/Conference — ICML 2025 poster_

### Official Review · Reviewer_6uKj · 2025-03-06

**Overall Recommendation:** 3

**Summary:**

* The paper introduces MonoAttn-Transducer, which enhances the handling of non-monotonic alignments in streaming generation by incorporating monotonic attention.
* The approach leverages the forward-backward algorithm to infer posterior alignment probabilities, enabling efficient training without enumerating exponentially large alignment spaces.
* Extensive experiments demonstrate that the proposed method significantly improves generation quality while maintaining comparable latency to baseline Transducer models.

**Claims And Evidence:**

* The effectiveness of MonoAttn-Transducer in handling non-monotonic alignments is demonstrated through experiments on speech-to-text and speech-to-speech simultaneous translation tasks.
* The paper provides detailed experimental results, including BLEU scores, COMET scores, and latency metrics (Average Lagging), showing consistent improvements over baseline Transducer models across various chunk size settings.
* Ablation studies and comparisons with prior alignment methods highlight the importance of learning monotonic attention through posterior alignment.

**Essential References Not Discussed:**

Since I am not familiar with the relevant literature, I cannot be sure.

**Experimental Designs Or Analyses:**

* The experiments include comprehensive comparisons with baseline models and state-of-the-art approaches.
* Results across different latency conditions (chunk sizes) provide insight into the flexibility and robustness of MonoAttn-Transducer.
* The evaluation on multiple datasets (MuST-C, CVSS-C) and different language pairs (En→De, En→Es) demonstrates the generalizability of the proposed approach.

**Methods And Evaluation Criteria:**

* The integration of monotonic attention into the Transducer architecture addresses the specific challenge of input-synchronous decoding limitations.
* The use of the forward-backward algorithm for posterior alignment inference maintains computational efficiency while enabling effective training.
* Evaluation on standard benchmark datasets (MuST-C, CVSS-C) and standard metrics (BLEU, COMET, Average Lagging) ensures that the results are comparable and relevant to the streaming generation research community.

**Other Comments Or Suggestions:**

* In some places, the text is a bit dense and could benefit from additional figures or diagrams to illustrate complex concepts.
* The discussion of related work could be more detailed, especially regarding how the proposed method differs from and improves upon existing approaches.

**Other Strengths And Weaknesses:**

The paper's main strength is its innovative approach to overcoming non-monotonicity in Transducer-based streaming generation by tightly integrating the predictor states with the source history through monotonic attention. This is a significant advancement as it improves the model's ability to handle complex tasks requiring non-monotonic alignments without a notable increase in latency. Furthermore, the method maintains the same time and space complexity as the baseline Transducer model, making it efficient and scalable.

However, the paper could be improved by providing more detailed theoretical analysis of why the proposed method works well, especially in comparison to other methods. Additionally, while the experimental results are convincing, more comprehensive ablation studies could strengthen the paper's claims.

**Questions For Authors:**

* How does the performance of MonoAttn-Transducer scale with larger datasets and more complex language pairs? (If the authors can provide additional experimental results or analysis on this, it would strengthen the paper's claims about the method's significance and applicability.)
* Could the proposed method be applied to other types of streaming generation tasks beyond speech translation, such as real-time text summarization or dialogue generation?
* What are the limitations of the current implementation, and what are the plans for future work to address these limitations?

**Relation To Broader Scientific Literature:**

* The paper builds upon previous work in Transducer models and monotonic attention, addressing their limitations in handling non-monotonic alignments.
* It contributes to the ongoing effort to improve the efficiency and effectiveness of streaming generation models, which is crucial for real-time applications such as simultaneous translation.
* The proposed method complements other approaches, such as attention-based encoder-decoder models and non-autoregressive models, offering a robust solution for complex streaming generation tasks.

**Theoretical Claims:**

The paper does not include formal proofs for theoretical claims. However, the theoretical foundations of the proposed method are based on established concepts:

* The use of the forward-backward algorithm for computing posterior probabilities is a well-known technique in hidden Markov models and sequence transduction.
* The formulation of monotonic attention is consistent with previous work in neural machine translation and speech recognition.

---

> ### Author Rebuttal · Authors · 2025-03-31
>
> **Thank you for your thoughtful review! We will make every effort to respond to your concerns.**
>
> >***1. However, the paper could be improved by providing more detailed theoretical analysis of why the proposed method works well, especially in comparison to other methods. Additionally, while the experimental results are convincing, more comprehensive ablation studies could strengthen the paper's claims.***
>
> Thank you for your suggestion. In the rebuttal period, we have designed an additional analysis to validate the rationale of our approach.
>
> The core technical aspect of this paper is dynamically computing the expected contextualized representation of monotonic attention through inference of the posterior alignment during training. This posterior alignment requires a prior alignment for calculation. **Naturally, there are two potential questions here: 1) Is posterior alignment absolutely necessary, or can the expected contextualized representation be computed directly using a prior alignment? 2) How does the choice of prior alignment affect the result of the posterior alignment, and is it robust?**
>
> For the first question, we attempted to directly calculate the expected contextualized representation using the prior probability, without inferring the posterior alignment. In Table 2, we present the results obtained by using a diagonal prior for this direct calculation. We found that this approach leads to a significant performance drop, particularly when the chunk size is small, where finer alignment is required. This addresses the first question. **Here, we will focus on the second question: the robustness of the posterior alignment.** We examine the impact of different prior choices: diagonal prior and uniform prior.
>
> |                           | **Chunk Size ($ms$)** | 320   | 640   | 960   | 1280  |
> |---------------------------|-----------------------|-------|-------|-------|-------|
> | **$p^{\mathrm{dia}}$**    | **BLEU**  | 24.72 | 26.74 | 27.05 | 27.41 |
> |                           | **AL ($ms$)**| 997   | 1239  | 1606  | 1991  |
> | **$p^{\mathrm{uni}}$**    | **BLEU**  | 24.89 | 26.68 | 27.26 | 27.11 |
> |                           | **AL ($ms$)**| 993   | 1249  | 1601  | 1983  |
>
>
> As shown, MonoAttn-Transducer's performance demonstrates robustness to the choice of prior alignment. **We visualize the posterior alignment when using different priors in this [annoymous URL](https://github.com/RandomUsername192/AnnoymousMaterials/blob/main/Comparison_cut.pdf). We have observed that, even with significant differences in the prior distribution, the posterior remains fairly robust.** This nice property reinforces the robustness of using the inferred posterior to train monotonic attention.
>
> >***2.Could the proposed method be applied to other types of streaming generation tasks beyond speech translation, such as real-time text summarization or dialogue generation?***
>
> Yes, the proposed method is a general framework that can be applied to many streaming generation task, as long as the target sequence consists of discrete tokens. We note a recent trend in research indicating that audio, images, and even video data can be tokenized into discrete sequences with minimal information loss [1,2]. Therefore, we believe this research has broad applicability to various real-time streaming tasks.
>
> [1] SpeechTokenizer: Unified Speech Tokenizer for Speech Large Language Models
> [2] Finite Scalar Quantization: VQ-VAE Made Simple
>
>
> >***3.How does the performance of MonoAttn-Transducer scale with larger datasets and more complex language pairs?***
>
> Thank you for your suggestion. We are currently working on scaling our approach with a larger dataset (GigaST) to further evaluate its performance. However, due to time constraints during the rebuttal period, the experiment has not been completed. We plan to include the results in the revised version.
>
>
> >***4.What are the limitations of the current implementation, and what are the plans for future work to address these limitations?***
>
> Currently, we have implemented the forward-backward algorithm and posterior inference using ***Numba***. However, we have observed that GPU usage during training occasionally is around 60%. We suspect that some inefficient operators in our implementation may be contributing to this. We believe that optimizing the implementation may benefit better scaling this approach.

---

### Official Review · Reviewer_MRT8 · 2025-03-09

**Overall Recommendation:** 3

**Summary:**

This paper investigates an interesting problem of non-monotonic alignment between input/output in streaming generation settings (e.g., simultaneous translation). The solution is to use the forward-backward algorithm to estimate alignment. Results show superior performance on speech-to-text and speech-to-speech simultaneous translation tasks compared with baseline techniques.

## Update after rebuttal

While I appreciate the contributions in this work, I share ntmp zJy7's concerns over the clarity of the methodology description. My question and the author's response made me realize that important pieces of technical details are missing. I would still like to keep my score, but would not argue for a accept for this reason.

**Claims And Evidence:**

Yes, the claims are reasonably supported by experimental evidence.

**Essential References Not Discussed:**

Would recommend adding a reference for forward/backward algorithm.

**Experimental Designs Or Analyses:**

Yes, the experimental design is reasonable. There is a small concern that this technique uses more compute than baseline (30% more), but I think this is largely acceptable given the notable improvements from baseline.

Results analysis also seems reasonable.

**Methods And Evaluation Criteria:**

Yes. The evaluation focuses on average latency, and standard machine translation metrics (BLEU, COMET), which seems befitting to the problem of speech to speech/text translation.

**Other Comments Or Suggestions:**

N/A

**Other Strengths And Weaknesses:**

This paper investigates a less-studied problem of non-monotonic alignment, which is a true problem in real-world applications. For the same reason, the paper is limited in that its impact is limited to these specific areas of real world problems.

**Questions For Authors:**

For Table 2., why would MonoAttn still out-perform Transducer with an infinite chunk size? I would anticipate that in such settings, alignment is not as important, as the context includes everything that may be relevant for genderating output?

**Relation To Broader Scientific Literature:**

This paper makes several connections to the brader literature.
- Extensive disucssion of transducer and its related work in Section 4.
- Extensive discussion of related prior arts in  Section 5.3, including Wait-k, RealTrans, etc...

**Theoretical Claims:**

N/A

---

> ### Author Rebuttal · Authors · 2025-03-31
>
> **Thank you for your acknowledgement of this work. We will make every effort to address your remaining concerns.**
>
> >***1. For Table 2., why would MonoAttn still out-perform Transducer with an infinite chunk size? I would anticipate that in such settings, alignment is not as important, as the context includes everything that may be relevant for genderating output?***
>
> Yes, when the chunk size is infinite, the posterior alignment would be that all predictor states align with the last encoder state with a probability of 1. This means that the context in the cross-attention would encompass the entire source. However, it is important to note that the naive Transducer does not include such an attention mechanism. Instead, its encoder and predictor are loosely coupled through a joiner (typically a simple MLP), which may limit its modeling capacity compared to our approach.
>
>
>
> >***2.This paper investigates a less-studied problem of non-monotonic alignment, which is a true problem in real-world applications. For the same reason, the paper is limited in that its impact is limited to these specific areas of real world problems.***
>
> Yes, the scope of this paper is limited to streaming sequence generation scenarios, specifically where the target sequence consists of discrete tokens. Given that an increasing number of studies have pointed out that data from audio, images and video modalities can be tokenized into discrete sequences in a manner similar to text [1,2], we believe this research can be widely applied to various real-time streaming tasks.
>
> [1] SpeechTokenizer: Unified Speech Tokenizer for Speech Large Language Models
> [2] Finite Scalar Quantization: VQ-VAE Made Simple
>
> >***3.Would recommend adding a reference for forward/backward algorithm.***
>
> Thank you for your suggestion! We will fix it.

---

### Official Review · Reviewer_ntmp · 2025-03-13

**Overall Recommendation:** 3

**Summary:**

This paper introduces MonoAttn-Transducer to tackle streaming generation tasks. MonoAttn-Transducer adds monotonic attention mechanism on top of Transducer and uses forward-backward algorithm to infer the alignment, which is then used to compute expected context representations used in monotonic attention. Experiments on simultaneous speech-to-text/speech translation (MuST-C and CVSS-C datasets) show that MonoAttn-Transducer performs better than vanilla Transducer when the chunk size of speech encoder is no less than 640ms.

## update after rebuttal

The authors have addressed my concerns regarding LAAL and the prior distribution. However, while they claim to have tested AlignAtt, no results were provided in the rebuttal. Additionally, I share Reviewer zJy7’s concern about the quality of the writing. Therefore, I will maintain my current score.

**Claims And Evidence:**

> Claim 1: vanilla Transducer have limited ability to attend to the input stream history during decoding, making it hard to manage reorderings. MonoAttn-Transducer manages re-ordering better with explicit monotonic attention.

This claim is true when the chunk size is large or the there is no more than 1 word reorder as shown in Figure 2. It is not true when the chunk size is small and there are more than 1 word reorders.

> Claim 2: an efficient training algorithm is proposed to avoid direct handling exponentially large number of translation trajectories.

This claim is true as shown in Table 2 and Section 6.2.

**Essential References Not Discussed:**

The paper claims to compare with previous state-of-the-art methods, but missing AlignAtt [1].

[1] Papi, S., Turchi, M., Negri, M. (2023) AlignAtt: Using Attention-based Audio-Translation Alignments as a Guide for Simultaneous Speech Translation. Proc. Interspeech 2023, 3974-3978, doi: 10.21437/Interspeech.2023-170

**Experimental Designs Or Analyses:**

Issues:
1. The author misses one important baseline AlignAtt [1], which is far better than EDAtt compared in Figure 2.
2. The AL in Table 3 is abnormal, both 118 and 153 ms are abnormally low, which I find hard to believe. One possible cause is over-generation.

[1] Papi, S., Turchi, M., Negri, M. (2023) AlignAtt: Using Attention-based Audio-Translation Alignments as a Guide for Simultaneous Speech Translation. Proc. Interspeech 2023, 3974-3978, doi: 10.21437/Interspeech.2023-170

**Methods And Evaluation Criteria:**

The method itself looks complicated at first glance, but the intuition behind is clear. One concern is equation (11). It is possible to obtain a better prior by either leveraging the confidence of pretrained MT models or word alignment methods as in [1]. Another concern is that the expected contextual representations still leads to training-inference mismatch.

The average lagging (AL) evaluation metric is problematic, since AL is not reliable if the over-generation happens. It is better to use length adaptive average lagging (LAAL) [2] as in recent IWSLT workshops [3].

[1] Wang, M., Vu, T. T., Wang, Y., Shareghi, E., & Haffari, G. (2024). Conversational simulmt: Efficient simultaneous translation with large language models. arXiv preprint arXiv:2402.10552.

[2] Sara Papi, Marco Gaido, Matteo Negri, and Marco Turchi. 2022. Over-Generation Cannot Be Rewarded: Length-Adaptive Average Lagging for Simultaneous Speech Translation. In Proceedings of the Third Workshop on Automatic Simultaneous Translation, pages 12–17, Online. Association for Computational Linguistics.

[3] Ibrahim Said Ahmad, Antonios Anastasopoulos, Ondřej Bojar, Claudia Borg, Marine Carpuat, Roldano Cattoni, Mauro Cettolo, William Chen, Qianqian Dong, Marcello Federico, Barry Haddow, Dávid Javorský, Mateusz Krubiński, Tsz Kin Lam, Xutai Ma, Prashant Mathur, Evgeny Matusov, Chandresh Maurya, John McCrae, Kenton Murray, Satoshi Nakamura, Matteo Negri, Jan Niehues, Xing Niu, Atul Kr. Ojha, John Ortega, Sara Papi, Peter Polák, Adam Pospíšil, Pavel Pecina, Elizabeth Salesky, Nivedita Sethiya, Balaram Sarkar, Jiatong Shi, Claytone Sikasote, Matthias Sperber, Sebastian Stüker, Katsuhito Sudoh, Brian Thompson, Alex Waibel, Shinji Watanabe, Patrick Wilken, Petr Zemánek, and Rodolfo Zevallos. 2024. FINDINGS OF THE IWSLT 2024 EVALUATION CAMPAIGN. In Proceedings of the 21st International Conference on Spoken Language Translation (IWSLT 2024), pages 1–11, Bangkok, Thailand (in-person and online). Association for Computational Linguistics.

**Other Comments Or Suggestions:**

When you mention the energy $e_{u,t}$, it is better to describe the meaning of it. Otherwise, it will be hard to understand for audience who did not read the monotonic attention paper.

**Other Strengths And Weaknesses:**

No.

**Questions For Authors:**

1. Could you please report the results with LAAL and include AlignAtt as the baseline? It will be interesting to see how MonoAttn-Transducer performs compared with a strong baseline under a more robust latency metric.
2. Could you please also report speech offset for simultaneous speech-to-speech translation?
3. Can you elaborate on why MonoAttn-Transducer does not improve over Transducer at chunk size of 320ms?

**Relation To Broader Scientific Literature:**

The primary contribution of this work is the integration of monotonic attention [1] into the Transducer framework, which overcomes the limitations of the standard Transducer in handling the word reordering challenges inherent in simultaneous translation.

[1] Naveen Arivazhagan, Colin Cherry, Wolfgang Macherey, Chung-Cheng Chiu, Semih Yavuz, Ruoming Pang, Wei Li, and Colin Raffel. 2019. Monotonic Infinite Lookback Attention for Simultaneous Machine Translation. In Proceedings of the 57th Annual Meeting of the Association for Computational Linguistics, pages 1313–1323, Florence, Italy. Association for Computational Linguistics.

**Theoretical Claims:**

No theoretical claims.

---

> ### Author Rebuttal · Authors · 2025-03-30
>
> **Thank you for your thoughtful review! We will make every effort to respond to your concerns.**
>
> >***1. Concerns regarding latency metric that are biased toward over-generation:*** *The AL in Table 3 is abnormal, both 118 and 153 ms are abnormally low, which I find hard to believe. One possible cause is over-generation. The average lagging (AL) evaluation metric is problematic, since AL is not reliable if the over-generation happens. It is better to use length adaptive average lagging (LAAL).*
>
> Thank you for your suggestion. Actually, we present the S2T results using LAAL in Table 6, App. C. Apologies for the confusion. For clarity, we have included the results from Table 6 here and added the S2S results using LAAL and Start Offset.
>
> **L: EN-ES R: EN-DE**
> | | **Chunk Size (ms)** | 320  | 640  | 960  | 1280 | 320  | 640  | 960  | 1280 |
> |-|-|-|-|-|-|-|-|-|-|
> | **Transducer** | **LAAL (ms)**|1168|1466|1847|2220|1258|1563|1942|2312|
> | |**LAAL_CA (ms)**|1381|1589|1944|2300|1444|1673|2028|2389|
> |**MonoAttn-Transducer**|**LAAL (ms)**|1230|1475|1837|2204|1317| 1582|1957|2305|
> | |**LAAL_CA (ms)**|1453|1607|1945|2295|1501|1702|2056|2387|
>
> **FR-EN S2S**
> ||**Chunk Size (ms)**|320|
> |-|-|-|
> | **Transducer**| **AL (ms)**|153|
> ||**LAAL (ms)**|984|
> ||**Start Offset (ms)**|1520|
> | **MonoAttn-Transducer**|**AL (ms)**|118|
> ||**LAAL (ms)**|918|
> ||**Start Offset (ms)**|1491|
>
> **We have realized that LAAL is a more robust metric for assessing latency (especially in S2S), and we will replace the metrics in the main body of the paper with LAAL for clearer presentation.**
>
>
> >***2.One concern is equation (11). It is possible to obtain a better prior by either leveraging the confidence of pretrained MT models or word alignment methods. Another concern is that the expected contextual representations still leads to training-inference mismatch.***
>
> Leveraging the confidence of pretrained MT models or word alignment methods for prior design is a possible approach. However, it will further complicate the training pipeline. In practice, we find that the posterior alignment is relatively robust to the choice of prior. Specifically, we examine the impact of different prior choices: diagonal prior and uniform prior.
>
>
> |                           | **Chunk Size ($ms$)** | 320   | 640   | 960   | 1280  |
> |---------------------------|-----------------------|-------|-------|-------|-------|
> | **$p^{\mathrm{dia}}$**    | **BLEU**  | 24.72 | 26.74 | 27.05 | 27.41 |
> |                           | **AL ($ms$)**| 997   | 1239  | 1606  | 1991  |
> | **$p^{\mathrm{uni}}$**    | **BLEU**  | 24.89 | 26.68 | 27.26 | 27.11 |
> |                           | **AL ($ms$)**| 993   | 1249  | 1601  | 1983  |
>
>
> As shown, MonoAttn-Transducer's performance demonstrates robustness to the choice of prior alignment. **We visualize the posterior alignment when using different priors in this [annoymous URL](https://github.com/RandomUsername192/AnnoymousMaterials/blob/main/Comparison_cut.pdf). We have observed that, even with significant differences in the prior distribution, the posterior remains fairly robust.** This nice property relieves concerns about the imperfect design of the prior.
>
>
> >***3.The author misses one important baseline AlignAtt [1], which is far better than EDAtt compared in Figure 2.***
>
> Thank you for the reminder. I have checked AlignAtt's paper and found that it performs particularly well at medium-to-high level latency. We will include it in Fig. 2 to facilitate a comparison between different streaming generation methods.
>
> >***4.Can you elaborate on why MonoAttn-Transducer does not improve over Transducer at chunk size of 320ms?***
>
> On one hand, when the chunk size is 320ms, the Transducer's prediction window is very small, which limits its flexibility in managing reordering. On the other hand, we found that the small improvement observed with a 320ms chunk size is mainly reflected when using BLEU as a metric. However, results on COMET show that MonoAttn-transducer still demonstrates significant improvement.
>
> >***5.Could you please also report speech offset for simultaneous speech-to-speech translation?***
>
> Thank you for your suggestion. We have incorporated the results into the table above.

---

### Official Review · Reviewer_zJy7 · 2025-03-14

**Overall Recommendation:** 2

**Summary:**

This paper proposes Mono-Attn-Transducer, a streaming sequence model that combines Transducer and monotonic attention. A novel training procedure that utilizes approximate alignment posteriors and alignment priors made training possible without expensive enumeration over an exponential search space or the use of a latency loss. Experimental results demonstrated strong performance on standard speech-to-text and speech-to-speech translation tasks.

## Update after rebuttal

Apologies for the delayed response. I have been sick the past 2 weeks.

Thanks for clarifying on the CAAT results! That addresses one of my main concerns with this paper. However, I am afraid the other concern (unclear presentation of the model architecture) is still valid. Without major revisions to the current draft, it would be very difficult for readers to grasp even just the high level architecture without looking at the code. For that reason, I could only raise my score to 2.

**Claims And Evidence:**

The main claim of this paper is the strong performance of the proposed method for streaming speech-to-text and speech-to-speech translation. To support this claim, this paper presents
-   Results on streaming speech-to-text translation in Table 2 and Figure 2
-   Results on streaming speech-to-speech translation in Table 3

The results were on standard datasets on these tasks, enabling comparison across a wide array of existing results.

However, I have a few concerns:
-   The results of CAAT in Figure 2 are quite a bit worse than what's reported in [Liu et al, 2021]. There, Tables 5 & 6 report the following BLEU/AL for En-De and En-Es for CAAT:

|En-De|AL(ms)|BLEU|
|-|-|-|
||508.1|20.5|
||813.8|21.4|
||1114.9|21.8|
||1443.4|22.2|
||1800.6|22.4|
||2137.8|22.6|
||Offline|23.2|

|En-Es|AL(ms)|BLEU|
|-|-|-|
||355.9|24.0|
||623.2|25.8|
||955.9|26.3|
||1275.9|26.4|
||1647.7|26.6|
||1977.3|27.1|
||Offline|27.5|

It's unclear whether the CAAT results in Figure 2 of this paper were reproduced by the authors themselves. What were reported in [Liu et al, 2021] are significantly better and much closer to results of Mono-Attn-Transducer in Table 2. If I am not mistaken, [Liu et al, 2021] used both smaller models and a smaller right context in their experiments. Thus a clarification on the discrepancy between the reported results would be extremely important for more reliably assessment on the actual contribution of Mono-Attn-Transducer.

-   The speech-to-speech translation results were only compared against the authors' own offline model, not even against [Zhao et al, 2024] which was cited in Section 5.4 (which reported much higher offline BLEUs). Admittedly this paper reported BLEUs under much lower latency than [Zhao et al, 2024], it would useful if more comparable results can be reported.

[Liu et al, 2021]: https://aclanthology.org/2021.emnlp-main.4.pdf
[Zhao et al, 2024]: https://arxiv.org/pdf/2410.03298

**Essential References Not Discussed:**

None.

**Experimental Designs Or Analyses:**

The experimental designs are sound and widely accepted practice.

**Methods And Evaluation Criteria:**

The proposed Mono-Attn-Transducer appears to be a sensible solution to streaming sequence prediction problems. However I would need some more information from the authors before I could make a good assessment, because several key details of the proposed method appear to be have been left out:
-   There is not a clear description of the overall architecture of Mono-Attn-Transducer. From the context, it appears that the overall architecture of Mono-Attn-Transducer is very similar to TAED in [Tang et al, 2023] except that previous decoder states are not updated when new source features become available. This however is only my best guess.
-   According to Algorithm 1, the training loss is the negative log-likelihood (from Equation (3)) based on $c_u$. $c_u$ is the approximate average cross attention output based on my interpretation of Equation (8). However $c_u$ depends on $s_u$, the predictor state which in turn depends on the actual alignment path $g(\cdot)$. It is not clear to me which alignment path is used to produce $s_u$.
-   Further, it is not clear to me how $c_u$ is used. My guess is that it's passed to the joiner to join with each $h_t$ to produce the Transducer sum of alignmenth path probabilities.

In summary, I am not confident that anyone with the information currently available in this paper can reproduce Mono-Attn-Transducer.

[Tang et al, 2023]: https://aclanthology.org/2023.acl-long.695.pdf

**Other Comments Or Suggestions:**

-   Euqation (1) is inaccurate and misleading for readers not familiar with Transducers. $\alpha(t, u)$ and $\beta(t, u)$ are marginal probabilities over alignment paths, not probabilities over the output prefix/suffix without blanks,
    -   $\alpha(t, u)$ is the sum of probabilities of partial alignment paths where $y_u$ being produced at $x_t$ is the last alignment label. $p(y_{1:u} | x_{1:t})$ is a confusing notation that strictly speaking includes probabilities of alignment paths where $y_u$ is produced anywhere between $x_1$ to $x_t$, followed by zero or more blanks.
    -   $\beta(t, u)$ is the sum of probabilities of partial alignment paths starting right after $y_u$ being produced at $x_t$. $p(y_{u+1:U} | x_{t:T})$ thus has the same problem as $p(y_{1:u} | x_{1:t})$. Additionally, $x_i$ are the source input features, instead of the encoder features $h_t$, so $\beta(t, u)$ depends on $x_{1:T}$ regardless of $t$ because $h_t$ in most unidirecitonal encoders can depend on the entire $x_{1:t}$, thus $\beta(t, u)$ depends on the entire $x_{1:T}$.
-   Equation (4) should probably include $y_{u-1}$ on the right hand side similar to Equation (5) of [Tang et al, 2023].
-   Section 3.2.1 should emphasize that the alignment posterior is approximate.
-   (230-235, left column): Bernoulli variables in monotonic attention and blanks in Transducer are mathematically equivalent. Standard Transducer simply folds the Bernoulli variable into a multi-class prediction. A variant of Transducer [Variani et al, 2020] actually uses a separate Bernoulli variable. Conversely, Transducer's dynamic programming algorithm can also be used to compute the marginals in monotonic attention (and is often one or two orders of magnitude faster than the so-called "efficient" algorithm in [Ma et al, 2023b]).

[Tang et al, 2023]: https://aclanthology.org/2023.acl-long.695.pdf
[Variani et al, 2020]: https://arxiv.org/pdf/2003.07705
[Ma et al, 2023b]: https://arxiv.org/abs/2312.04515

**Other Strengths And Weaknesses:**

Strengths
-   The novel training method is effective and elegant in not needing a latency loss.
-   The reported speech-to-text translation results are strong even though the comparison with CAAT/TAED is inconclusive.

Weaknesses
-   Key details are missing that prevent the readers from fully understanding or reproducing Mono-Attn-Transducer.

**Questions For Authors:**

As outlined above, my main concerns are the comparison with CAAT, and the lack of key architecture details.

**Relation To Broader Scientific Literature:**

This paper is a continuation in the exploration of enhancing a Transducer model with cross attention in streaming sequence prediction. CAAT ([Liu et al, 2021]) and TAED ([Tang et al, 2023]) are the two existing papers most closely related to this paper. The key technique in this paper draws inspiration from the line of work on monotonic attention (such as [Raffel et al, 2017] and [Arivazhagan et al, 2019])
-   CAAT separates self attention and cross attention layers in the Transformer decoder, using the stack of self attention layers as the Transducer predictor to encode the output history without any alignment information, and the stack of cross attention layers as the joiner. As a result, the standard dynamic programming algorithm for computing the sum of alignment path probabilities in a Transducer lattice is still applicable. Because waiting until the end of source gives the joiner the most complete source information, a latency loss is necessary in CAAT training to prevent degenerate model behavior.
-   TAED took a different path and kept the Transformer decoder intact as the Transducer predictor. At each time step $t$, the Transformer decoder states are completely recomputed with all the source features available so far ($h_{1:t}$). As a result, the standard dynamic programming algorithm for computing the sum of alignment path probabilities in a Transducer lattice is also applicable at the cost additional Transformer decoder computation. An AED loss term is introduced to, ostensibly, prevent degenerate model behavior similar to CAAT.
-   Mono-Attn-Transducer took the average cross attention feature technique commonly employed in monotonic attention ([Raffel et al, 2017] and [Arivazhagan et al, 2019]) to approximate the sum of alignment path probabilities, with a novel alignment prior mechanism and a two-stage estimation. The strong alignment priors ensure that training would not lead to degenerate model behavior.

[Raffel et al, 2017]: https://proceedings.mlr.press/v70/raffel17a/raffel17a.pdf
[Arivazhagan et al, 2019]: https://arxiv.org/pdf/1906.05218
[Tang et al, 2023]: https://aclanthology.org/2023.acl-long.695.pdf
[Liu et al, 2021]: https://aclanthology.org/2021.emnlp-main.4.pdf

**Theoretical Claims:**

There are no theoretical claims in this paper.

---

> ### Author Rebuttal · Authors · 2025-03-30
>
> **Thank you for your efforts in reviewing! We will make every effort to respond to your concerns.**
>
> >***1.However I would need some more information from the authors before I could make a good assessment.***
>
> **We will address your questions about the method point by point.**
>
> 1. >*It appears that the overall architecture of Mono-Attn-Transducer is very similar to TAED except that previous decoder states are not updated when new source features become available. This however is only my best guess.*
>
>     Yes, in the inference of Mono-Attn-Transducer, the generated predictor states are not updated when receiving new source features. When the predictor encodes the $u$-th target state, it depends on previous predictor states and the currently available source:
>
>     $s_{u} = f_{\theta}(s_{0:u-1},h_{1:g(u)},y_{u-1})$
>
> 2. >*$c_u$ is the approximate average cross attention output based on my interpretation of Equation (8). However $c_u$ depends on $s_u$, the predictor state which in turn depends on the actual alignment path $g(\cdot)$. It is not clear to me which alignment path is used to produce $s_u$.*
>
>    In fact, the core idea of the approach lies in the uncertainty of the alignment path used to produce $s_u$ in training. Therefore, a posterior approximation is required to guide the learning process. To analyze this issue, we can further examine Eq. 4 in detail. In fact, the predictor producing $s_u$ primarily relies on two steps:
>
>     A. **Self-attention** with earlier predictor states ($s_{0:u-1}$): **This is deterministic.**
>
>     B. **Cross-attention** with encoder states ($h_{1:g(u)}$): Since the specific value of $g(u)$ is unknown in training, **we instead estimate the posterior alignment (Eq. 7) and use the resulting alignment probability to approximate the expected context representation $c_u$ in the cross-attention (Eq. 8).**
>
> 3. >*Further, it is not clear to me how $c_u$ is used. My guess is that it's passed to the joiner to join with each $h_t$ to produce the Transducer sum of alignment path probabilities.*
>
>    As mentioned earlier, $c_u$ represents the expected context representation (**in other words, $c_u$ is the output of the cross-attention module**). This $c_u$ is then passed to the next module in the predictor, typically an FFN. **We refer to the final output of the predictor as $s_u$** (Eq. 4), which is subsequently passed to the joiner.
>
> >***2.Key details are missing that prevent the readers from fully understanding or reproducing Mono-Attn-Transducer.***
>
> **We hope the previous clarifications help better understanding. We also provide the source code in this [anonymous URL](https://github.com/RandomUsername192/AnnoymousMaterials) to ensure reproducibility.**
>
>
> >***3.Concerns on Baseline Results***
>
> 1. >*Clarification of CAAT results.*
>
>    We used the code provided by CAAT for replication but did not obtain the results reported in their paper. Since CAAT did not provide the distillation data used to train their model, we trained with the same data we used for training the Transducer, which could be the source of the discrepancy. The official CAAT config (audio_cat) does use fewer params; however, we argue that this is a compromise made due to its $O(T)$ memory usage in training.
>
> 2. >*Clarification of S2S results.*
>
>     In fact, Zhao et. al specifically augmented the speech synthesis module, while the Transducer remains unchanged. They use an acoustic LM to refine semantic tokens generated by Transducer and then generate the waveform. **However, their design is not directly related to the Transducer itself, but rather to improving waveform generation.** Our focus is on the Transducer, and we opt for a standard approach [1]: directly converting generated semantic tokens to waveforms using a vocoder.
>
>     [1] Speech Resynthesis from Discrete Disentangled Self-Supervised Representations
>
>
>  >***4.Comments on definition of $\alpha(t,u)$ and $\beta(t,u)$.***
>
>    We noticed you argued that $\alpha(t,u)$ is the sum of probabilities of partial alignments where $y_u$ being produced at $x_t$ is the last alignment label.
>
> **However, this is not correct. $\alpha(t,u)$ does incorporate the probabilities where $y_u$ is followed by zero or more blanks. $\alpha(t,u)$ is precisely defined as the probability of generating $y_{1:u}$ from $x_{1:t}$ in the original Transducer paper (see Section 2.4 of the paper). Similarly, $\beta(t,u)$ is the probability of outputting $y_{u+1,U}$ from $x_{t:T}$. There is no need for $y_u$ being produced at $x_t$**.
>
> >***5.Suggestions on Equation (4) and Section 3.2.1.***
>
> Thank you! We will fix it.
>
> >***6.Comments on Equivalence between monotonic attention and Transducer.***
>
> We quite agree with this statement. Our approach actually leverages the DP algorithm of the Transducer to assist in the computation of monotonic attention. We will include Variani et al. (2020) in the discussion of the paper. Thank you for the reminder.

---

### Decision · Program_Chairs · 2025-05-01

**Decision:**

Accept (poster)

**Comment:**

This paper proposes Mono-Attn-Transducer, a streaming sequence model to address tasks requiring non-monotonic alignments, such as simultaneous translation. Three reviewers give positive ratings while one reviewer gives negative ratings with some questions, such as more information about the method is needed before making a good assessment, and key details are missing that prevent the readers from fully understanding or reproducing Mono-Attn-Transducer. While the authors have provided adequate information to answer the author's questions, the reviewer did not respond further. Overall, I think the response provided by the author is informative to address the concerns to some extent. Therefore, I recommend an acceptance to this paper.